# Snails In Silico: A Review of Computational Studies on the Conopeptides

**DOI:** 10.3390/md17030145

**Published:** 2019-03-01

**Authors:** Rachael A. Mansbach, Timothy Travers, Benjamin H. McMahon, Jeanne M. Fair, S. Gnanakaran

**Affiliations:** 1Theoretical Biology and Biophysics Group, Los Alamos National Laboratory, Los Alamos, NM 87545, USA; mansbach@lanl.gov (R.A.M.); tstravers@lanl.gov (T.T.); mcmahon@lanl.gov (B.H.M.); 2Center for Nonlinear Studies, Los Alamos National Laboratory, Los Alamos, NM 87545, USA; 3Biosecurity and Public Health Group, Los Alamos National Laboratory, Los Alamos, NM 87545, USA; jmfair@lanl.gov

**Keywords:** conotoxins, conopeptides, computational studies, molecular dynamics, machine learning, docking, review, drug design, ion channels

## Abstract

Marine cone snails are carnivorous gastropods that use peptide toxins called conopeptides both as a defense mechanism and as a means to immobilize and kill their prey. These peptide toxins exhibit a large chemical diversity that enables exquisite specificity and potency for target receptor proteins. This diversity arises in terms of variations both in amino acid sequence and length, and in posttranslational modifications, particularly the formation of multiple disulfide linkages. Most of the functionally characterized conopeptides target ion channels of animal nervous systems, which has led to research on their therapeutic applications. Many facets of the underlying molecular mechanisms responsible for the specificity and virulence of conopeptides, however, remain poorly understood. In this review, we will explore the chemical diversity of conopeptides from a computational perspective. First, we discuss current approaches used for classifying conopeptides. Next, we review different computational strategies that have been applied to understanding and predicting their structure and function, from machine learning techniques for predictive classification to docking studies and molecular dynamics simulations for molecular-level understanding. We then review recent novel computational approaches for rapid high-throughput screening and chemical design of conopeptides for particular applications. We close with an assessment of the state of the field, emphasizing important questions for future lines of inquiry.

## 1. Introduction

Marine cone snails from the family *Conidae* capture their prey and defend themselves using venoms containing short proteins called conopeptides [1,2]. The majority of these toxins range in sequence length from 10 to 45 amino acids, with a median size of 26 residues [3]. Every species from the family *Conidae* can produce in excess of a thousand types of conopeptides; it is estimated that that only 5% of the peptides are shared between different species [4]. This large chemical diversity is primarily driven by evolutionary pressure for improving defense and/or prey capture [2], with sudden ecological changes likely driving the selection of new fast-acting conopeptides [5,6]. Although several classes of “disulfide-poor” conopeptides have been recently identified [7,8], the majority of cone snail toxins contain multiple disulfide linkages within a single peptide chain that allow the adoption of highly-ordered structures [9]. In fact, disulfide bond formation is the most prevalent type of posttranslational modification seen in conopeptides [10], although other types of modifications have also been observed, including proline hydroxylation [11], tyrosine sulfation [12], C-terminal amidation [13], O-glycosylation [14], and addition of gamma-carboxyglutamic acid [15].

During the review of the current literature on conopeptides, we noticed that the term “conotoxin” has sometimes been used interchangeably with the term “conopeptide” [15,16]. In this review, following the definition given in [17], we instead draw a distinction and employ the term “conotoxin” to refer to the specific subset of the conopeptides that contain two or more disulfide bonds.

Conopeptides are potent pharmacological agents that bind with high specificity to their target proteins (equilibrium dissociation constants or $k_{D}$ values in the nM range) [18]. Broadly, the protein families targeted by conopeptides are grouped into the following three categories [19]: (i) ligand-gated channels such as nicotinic acetylcholine receptors (nAChRs) [20]; (ii) voltage-gated channels for sodium [21], potassium [22], and calcium [23]; and (iii) G protein-coupled receptors (GPCRs) [24]. Although these targets belong to various protein families, the same physiological effect is achieved by conopeptide binding: disruption of signaling pathways, which leads to the inhibition of neuromuscular transmission and, ultimately, prey immobilization [25,26].

Due to their highly specific and potent binding modes, conopeptides can exhibit significant toxicity in humans—*Conus geographus* stings have reported fatality rates of 65 percent—which has led to discussions of weaponization potential by biosecurity experts and establishment of USA federal regulations that place restrictions on research into particular conopeptide classes [27,28,29]. Nevertheless, the conopeptide chemical space is vast and most are not considered to be bioterrorism threats; indeed, conopeptides have become useful research tools for understanding the physiological functions of their target proteins and have emerged as valuable templates for rational drug design of new therapeutic agents in pain management [30,31,32,33,34,35,36]. An important milestone was the approval of the conotoxin ω-MVIIA from *Conus magus* as a commercial drug for chronic pain under the name Prialt (generic name ziconotide) [37,38].

Recent years have seen a growing availability and refinement of computational resources and algorithms that can be used for gaining more insights on structure-function relationships in conopeptides. For instance, there is now an increasing emphasis on the use of in silico methods, either alone or in combination with experimental techniques, for molecular-level understanding and protein engineering for drug design [39,40]. The explosion of machine learning (ML) techniques and use-cases has led to a focus on the creation of large databases that can be mined for predictions [41]. Meanwhile, molecular dynamics simulations offer a straightforward and ever-more-efficient method for probing protein conformations in detail [42,43,44], while docking studies provide a rapid complementary method to predict binding affinities and modes of ligands bound to large complexes [45,46]. Finally, combinations of these methods are being applied to design problems in such disparate areas as the creation of drug-like molecules [47], the identification of antimicrobial peptides [48], and the discovery of novel materials [49].

In this review, we provide an overview of how such computational techniques have been exploited to enrich our understanding of the molecular mechanisms behind conopeptide function and binding, predict their targets and binding affinities, and ultimately design novel conopeptides for specific applications in a rational manner. We begin in Section 2 by discussing the classification and structure of the family of conopeptides in general. In Section 3, we present a detailed review of computational studies that have been performed, ranging from machine learning predictors of conopeptide categories to molecular dynamics and docking studies for structure and folding characterization and binding mode elucidation to large-scale computationally-driven rational design of conopeptides for specific applications. In Section 4, we discuss the current state of the field, and, in Section 5, we briefly conclude.

## 2. Background

Despite the vast number of different conopeptides and their overall diversity, some similarities do emerge, and it is customary to employ different classification schemes to categorize information about different aspects of these similarities. Since such similarities may arise from different origins—for example, evolutionary similarity versus functional similarity—classification schemes encoding different information do not necessarily provide overlapping categories. In the next sections, we introduce some of the most commonly used schemes and discuss their relationships and some implications for understanding the structures and functions of conopeptides.

### 2.1. Categories of Conopeptide Classification

Classification schemes exist that describe aspects of conopeptide similarity ranging from those based solely on sequence to those based on a mixture of sequence and structural properties to those based on specific in vivo functionality [3,50], some of which are simple to determine for all conopeptides with known sequence, others of which have only been determined for small subsets of all identified conopeptide sequences (see Table 1 for a summary).

#### 2.1.1. Gene Superfamily

Gene superfamily is an evolutionary classification in which conopeptides are assigned a category based on clustering of the slowly evolving region of the precursor protein that is processed by the endoplasmic reticulum [1]. Gene superfamily is strictly a mark of evolutionary similarity within the precursor regions, and is of limited use in finding conservation patterns within the hypervariable regions that are actually transcoded into mature toxins. (See Figure 1 for an example of two conopeptides of gene superfamily A that have highly similar precursor regions but very different mature toxin sequences.)

#### 2.1.2. Cysteine Framework and Loop Class

In contrast to the gene superfamily, the cysteine framework of a conopeptide is a category assigned based on the sequence of amino acids of the toxin region itself. It generally refers to the pattern of neighboring and non-neighboring cysteines in the sequence [50] (It should be noted that there has historically been some confusion in the literature over the use of the term “cysteine framework.” In addition to the way it is presented here, it has sometimes been used to refer to a more structure-based categorization that includes the pattern of disulfide bonds that form between non-neighboring cysteines. We describe disulfide connectivity separately and adopt the definition of cysteine framework as described by [50]). For example, Figure 2 shows examples of framework I, corresponding to the cysteine pattern CC-C-C, and framework III, corresponding to the cysteine pattern CC-C-C-CC. There is one exception to the general rule for framework definition: peptides with a cysteine pattern of CC-C-C and a hydroxyproline residue between cysteines three and four are assigned to framework X, while all other peptides with a cysteine pattern of CC-C-C are assigned to framework I. Twenty-seven such frameworks are currently recognized (see Table 2), and any new conopeptide may be straightforwardly assigned a framework from knowledge of its amino acid sequence. If, in addition to noting which cysteines are neighboring and which are not, one chooses to classify conopeptides based on intrasequence cysteine distance, one may employ the conopeptide loop class instead of the cysteine framework. A particular loop class is defined by the number of amino acids between the cysteines, where that number is zero for neighboring cysteines (see Figure 2 for two examples).

#### 2.1.3. Fold and Subfold Class

If instead of employing a sequence-based classification, one employs a structure-based scheme, conopeptides may also be divided into different fold and subfold classes due to the high structural similarities that are often enforced by the disulfide connectivities of the cysteines (see also Section 2.1.4 and Section 2.2.2). Fold and subfold are structural categorizations for conopeptides that have determined three-dimensional (i.e., secondary and tertiary) structures [3,55], with subfold being a subset of fold class that encompasses finer secondary structural detail. There are 13 folds and 18 subfolds currently defined (see Figure 3), but at the time of this article there are only 161 determined conopeptide structures in the Protein Data Bank (PDB), of which 114 represent unique sequences or post-translationally modified sequences [56], and many fall into only four of the thirteen folds: A, B, C, and D (cf. Figure 3a–d and Figure 6a). For a more detailed discussion of the precise characteristics of each fold and subfold class, the interested reader is referred to Akondi et al. [3].

#### 2.1.4. Disulfide Connectivity

One aspect of conopeptides that has sometimes been employed for their classification is the disulfide connectivity of their cysteines; that is, which cysteines are connected to which via covalent disulfide bonds. Disulfide connectivity is of particular interest for two reasons: (i) patterns of disulfide connectivity in the primary structure of conopeptides play an important role in defining their three-dimensional structure [69] and thus their fold/subfold classes, although the extent to which specific disulfide bonds are important to retaining the native structure can be peptide-dependent [70], and (ii) although it is common to describe conopeptides in terms of their “native” disulfide connectivity, multiple different connectivities for a single peptide have been observed in vitro, which may correspond to stable, metastable, or off-pathway structural isomers, and which often display different properties than the native structure [71] (see Section 2.2.2 and Section 4 for further discussion of this point).

In Figure 4, we show the structures resulting from some common disulfide connectivities. For instance, in conopeptides containing four cysteine residues, disulfide formation between cysteines 1–3 and 2–4 leads to the so-called “globular” structure with α-helical content (Figure 4a left and Figure 3a). In contrast, disulfide formation between cysteines 1–4 and 2–3 leads to the so-called “ribbon” structure with β-sheet content (Figure 4a right and Figure 3d). As another example, in conopeptides containing six cysteine residues, disulfide formation between cysteines 1–4, 2–5, and 3–6 leads to a “cysteine knot” structure (Figure 4b left and Figure 3c). However, disulfide formation between cysteines 1–5, 2–4, and 3–6 leads to a different overall tertiary structure (Figure 4b right and Figure 3h).

#### 2.1.5. Pharmacological Family

Finally, if one chooses to classify conopeptides based on their mode of action, their pharmacological family may be assigned. Pharmacological families group conopeptides based on their target receptor and the type of interaction with this receptor, of which 12 such families are currently recognized [17,54] (see Table 3); however, mode of action is also not a quantity that is straightforward to determine from the sequence, and as of now, only 243 of the more than 6000 sequences downloadable from the Conoserver (an automatically-updating online repository of information about conopeptides [17,54]) have an associated pharmacological family (The Conoserver is the most complete repository of compiled information on the conopeptides the authors of this review were able to find and forms an excellent reference for any researcher in the field; however, it should be noted that there are over 400 sequences that currently appear to be inaccessible via the search function or download. Readers may refer to the Uniprot database as well [75].).

### 2.2. Relationships between Categories

Knowing how a conopeptide is categorized under one classification scheme is not necessarily indicative of its categorization under another. For example, conopeptides from a particular cysteine framework are not limited to belonging to a single pharmacological family. Conopeptides in framework I can target either nicotinic acetylcholine receptors (nAChRs) or adrenoceptors, while those in framework III can target either nAChRs or sodium/potassium channels. Despite a generally demonstrated selectivity for their targets, certain peptides even display binding affinity for more than one receptor type [76,77]. Nonetheless, there may be similarities between the ways in which different classification schemes group the conopeptides (see Figure 5).

#### 2.2.1. Cysteine Framework, Loop Class, Fold, and Pharmacological Family

To be more specific, in Figure 5a, we demonstrate the similarities and differences between the ways in which conopeptides are grouped when using cysteine framework versus pharmacological family as the classification. We demonstrate groupings for all 243 peptides with defined pharmacological families downloadable from the Conoserver, comprising 137 sequences with four cysteines, 99 sequences with six cysteines, two sequences with eight cysteines, and five sequences with ten cysteines; the image is a visualization of a pairwise matrix of the considered conopeptide dataset. The overall dataset demonstrates that the categories overlap quite strongly, both visually and numerically: 85% of peptide pairs with the same cysteine framework have the same pharmacological target and 88% of peptide pairs with different frameworks have different targets. If we eliminate all frameworks containing four or fewer cysteines (frameworks I and X, comprising about half the dataset, see Figure 5b), however, then only 45% of peptide pairs with the same cysteine framework have the same pharmacological target, although 93% of peptide pairs with different frameworks have different targets. This precipitous drop in categorical overlap for the same frameworks implies that target prediction becomes more difficult as the number of cysteines in the sequence increases, which may be a consequence of the rapidly growing number of possible disulfide bond connectivities for systems with more than four cysteines (see also Section 4 for a brief discussion of this point).

In Figure 5c,d, we demonstrate the similarities and differences between conopeptide classification as described by the (c) fold and (d) subfold and pharmacological family categories for a subset of the peptides described in Akondi et al. [3]. We focus only on cysteine-containing peptides that have a known pharmacological family and a defined fold and subfold, which results in a set of 80 peptides comprised of one sequence with two cysteines, 45 sequences with four cysteines, 32 sequences with six cysteines, and two sequences with eight cysteines. We see that there is a relatively large overlap between fold class and pharmacological target: 76% of peptide pairs that are assigned the same fold class have the same pharmacological target, and 89% of peptide pairs that are assigned a different fold class have different targets, while 80% of peptide pairs assigned the same subfold class have the same pharmacological target and 83% of peptide pairs assigned different subfold classes have different targets. It should be noted, however, that this is a small subsample of the vast array of possible conopeptides and that bias in the assessed peptides may lead to artificially high cross-categorical similarity due to high sequence similarity that would not be expected in general, particularly since there are no sequences in the dataset containing more than eight cysteines and most contain six or fewer. Indeed, almost 75% of all conopeptide studies have been performed on the four-cysteine α-conotoxins [78].

In Figure 6, we demonstrate the extent to which loop class can discriminate between the groupings of other categorization schemes, using the 103 cysteine-containing peptides from Akondi et al. [3]. The latter dataset comprises seven sequences with two cysteines, 59 sequences with four cysteines, 35 sequences with six cysteines, and two sequences with eight cysteines. The distance between a pair of points in this plot is a visual indication of the difference between their loop classes, while different colors represent different categories. If loop class is indicative of a category, then points with the same color should be clustered together, whereas if loop class and the other classification type have no relation, then nearby points should appear to be colored randomly. Panel (a) of Figure 6, in which different colors represent different folds, demonstrates several clusters that are primarily one color, with the major sources of overlap between categories being the central cluster of points where folds A, D, and E reside almost on top of one another. Indeed, there are several overlapping points with identical loop classes that have different folds (see also Section 2.2.2).

Panel (b) of Figure 6, in which different colors represent different pharmacological families, also demonstrates some clustering of similar colors, although there are several points that have the same color as their neighbors in panel (a) but different colors in panel (b) (see red arrows for three examples). Thus, there is an overlap between pharmacological family and loop class, but it is somewhat weaker than that of the overlap between fold and loop class, even in this small dataset. This demonstrates that pharmacological family is not solely dictated by three-dimensional structure (see also Section 3.2 for a detailed discussion of molecular mechanisms that control selectivity and binding affinities for target receptors).

#### 2.2.2. Disulfide Connectivity Determines Fold

Another important point brought up by Figure 6 is the extent to which fold is dictated by disulfide connectivity over detailed sequence. The blue arrows indicate an example where an identical sequence has a very different structural fold and may also have a different target (the isomer’s target is often unknown). We elaborate a more specific set of examples in Figure 7a. Conotoxins α-GI [60,81] and α-BuIA [82,83] have a native connectivity between cysteines 1–3 and 2–4 that corresponds to the “globular” motif, but altering this connectivity to cysteines 1–4 and 2–3 forces both to adopt a “ribbon” motif. Interestingly, the disulfide connectivity has also been found to impact the conformational dynamics of the conopeptide. In the case of conotoxin α-GI, the native “globular” conformation was found to have less backbone conformational variability than the induced “ribbon” conformation [81]. The reverse is true, however, for conotoxin α-BuIA, where the induced “ribbon” motif was found to have less backbone conformational variability than the native “globular” motif, even though only the latter motif is capable of inhibiting the function of the target nAChR [83].

Not only do different disulfide connectivities imply different structures irrespective of sequence, but, perhaps more importantly, similar disulfide connectivities imply similar structure irrespective of sequence, as demonstrated in Figure 7b. The conotoxins α-EI (PDB 1K64 [84]) and α-IMI (PDB structure 1G2G [51]), for instance, both adopt a “globular” connectivity with disulfide linkages between cysteines 1-3 and 2-4 (Figure 7b top). However, a sequence alignment between these two conotoxins shows no sequence identity or similarity besides a single proline and the four cysteine residues (Figure 7b bottom). Surprisingly, even diverse sequences with a different number of disulfide bonds have been found to adopt similar structural folds if similar underlying disulfide connectivities exist. For example, the conotoxins ω-MVIIA (PDB 1MVJ [85]) and ι-RXIA (PDB 2P4L [86]), which have six and eight cysteines respectively, have both been found to adopt a cysteine knot. This structural motif is the most commonly observed in small disulfide-rich proteins, occurring in nearly 40% of available structures [55]. It is comprised of a ring formed by two of the disulfide bonds and the interconnecting backbone, with the third disulfide bond passing through this ring [87]. In conotoxin ι-RXIA, which has eight cysteines, the cysteine knot arises from disulfide bond formation between cysteines 1–4, 2–6, and 3–7, with cysteines 6 and 7 taking the place of cysteines 5 and 6 in a six-cysteine conotoxin (see e.g., of conotoxin μ-GS in Figure 4b) [86].

## 3. Computational Strategies to Understand and Predict Conopeptide Structure and Function

The increased and efficient discovery of new conopeptides has been facilitated by advances in the fields of transcriptomics, proteomics, and bioinformatics and their integration into a new field called venomics, the in-depth study of venoms [88,89,90]. As we have demonstrated, the understanding of conopeptides is complicated by their diversity along many different axes, making application-specific design a difficult if not intractable problem. In this section, we discuss different ways in which in silico methods have been used to partially address these questions, both in terms of detailed characterization of individual conopeptides and in terms of identification of broad conopeptide trends (see Figure 8 for a visual summary of methologies and Table A1 for a tabulated version of the references cited in this section). We begin by introducing in Section 3.1 the progress that has been made in employing machine learning techniques to predict the pharmacological targets of conopeptides from their sequences. Then, in Section 3.2, we report the ways in which docking studies and molecular dynamics simulations have been used—often in conjunction—to shed light on the structure and function of a number of individual conopeptides and conopeptide pharmacological families. Finally, in Section 3.3, we provide an overview of computationally-driven studies that were used to design conopeptides for specific applications.

### 3.1. Predicting Function from Sequence through Machine Learning

Classification via data-mining of a set of training data encompasses a broad set of techniques that have been growing in popularity in recent years, demonstrating utility in such disparate areas as agriculture [91], medical image processing [92], bioinformatics [93], and many others. Attempts have been made to employ such methods to the categorization of the incredibly diverse group that makes up the conopeptides. Indeed, extensive work has been done on developing sequence-based predictors that use machine learning techniques for identifying the target receptor type of a novel conopeptide sequence, achieving predictive accuracies of up to 97% [94], although they are currently limited to discriminating the toxins targeting voltage-gated sodium, calcium, and potassium channels. Additionally, a very recent tool, ConusPipe, has attempted the classification of RNA sequences by employing three different machine learning models, in order to identify potential novel conotoxin sequences within *Conus* transcriptomes without resorting to sequence homology type searches, which may fail due to the high diversity of the sequences in question [95].

A number of different peptide representations and classification algorithms have been experimented with to optimize performance. Innovations include representing conopeptide sequences by their amino acid composition and dipeptide composition [96]—that is, the composition based on neighboring amino acid pairs—or their “pseudo” amino acid composition, which incorporates information about the correlation between physicochemical properties of the compositional amino acids [97]. Dimensionality reduction to identify pertinent features has improved performance: relatively recent feature selection techniques employed include the binomial distribution [98], the relief algorithm [99], the f-score algorithm [97,100], and analysis of variance [96]. Several relatively well-known machine-learning algorithms have demonstrated good performance: support vector machines [96,97,100], radial basis functions [98], and random forests [99]. The best performance was achieved by ICTCPred, with an average accuracy of 0.973, an overall accuracy of 0.957, and a minimum sensitivity of 0.919 for correctly discriminating between the possible targets of voltage-gated sodium, potassium, or calcium channel on a testing dataset of 70 conopeptides [99] (It should be noted that a manual investigation of the training set for this study suggests it was not properly pruned, as several ion channel sequences were erroneously included and labeled as conopeptides; however, this minor error did not appear to have a strong impact on the results, as the testing set did not include any such errors.). For a more detailed explanation, we refer the interested reader to a comprehensive review by Dao et al. [94] of recent machine learning techniques as applied to the conopeptides.

Despite the availability of algorithms that can accurately predict target receptors based on conopeptide sequence [94,99,101] with high accuracy, these methods are still subject to a number of limitations. The size of current benchmarks for the conopeptides is not large enough to imply generalizability over the entire family. For example, we found 243 conopeptides in the Conoserver with identified pharmacological targets, out of a total of 6254 identified sequences [17,54], while most predictor studies have used training sets on the order of about 100–150 sequences to eliminate redundancy and reserve data for testing [94]. The high variability of conopeptide sequence, in addition to the possibility of synthetic analogues, suggests that this is not sufficient to create a classifier that would be accurate for a general sequence. Furthermore, these methods do not provide details on the mechanisms of how conopeptides function upon binding to their targets. Such details can be elucidated mainly through modeling and analysis of conopeptide structures, which can allow for the estimation of folding free energies and binding affinities. In addition, the relative toxicity of a conopeptide in different target animals can be more reliably assessed from a structural perspective. Finally, understanding how rational modifications in engineered conopeptides can alter their toxicity in specific animals, including humans, necessitates a more mechanistic approach, which is the focus of the next few sections.

### 3.2. Docking Studies and Molecular Dynamics Simulations for Understanding of Conopeptide Structure and Binding

Docking studies and molecular dynamics (MD) simulations are complementary computational techniques used to study proteins and peptides in molecular-level detail. Docking studies provide an in silico representation of the binding between a receptor and its ligand, which may be used for high-throughput screening and prediction of binding energies and affinities [45]. The overall procedure consists of a search through the conformational space of the “docked” complex followed by scoring of the possible configurations thus identified. Although there are still some outstanding questions in the field, including the optimal method for handling solvation, how to model flexibility of the target binding site, and how to correctly assess protein-protein interactions when the ligands are not small molecules, docking studies nonetheless boast a rich history of profitable application in a number of areas [102]. Molecular dynamics simulations, which typically consist of solving Newton’s laws of motion over a number of time steps, can complement docking for the understanding and prediction of molecular mechanisms, as well as being a sophisticated technique in their own right. They can better handle conformational entropy and predict dynamical fluctuations at the cost of significantly higher computational expense. In a sense, if docking studies provide breadth of understanding of protein systems, MD simulations provide depth. For example, an attractive procedure for designing compounds to interact with a particular target is to perform docking to identify an initial set of potential compounds followed by performing MD for their detailed characterization [103,104].

With respect to the conopeptides, docking studies and/or MD simulations have been used to (i) complement nuclear magnetic resonance (NMR) and X-ray crystallography experiments to characterize the three-dimensional structure of conopeptides and receptor/toxin complexes, (ii) investigate the importance of structure, electrostatics, and hydrophobicity for receptor/toxin binding and identify key residues contributing to binding, (iii) identify and characterize molecular mechanisms of binding and binding/unbinding pathways, and in a few cases (iv) to characterize thermodynamics, kinetics, and environmental effects on conopeptide folding.

#### 3.2.1. Conopeptide and Receptor Structural Characterization

Structure is an important predictor of function: conopeptides are, broadly speaking, either steric pore blockers or “lock-and-key” (ant)agonists that bind to and alter the functional properties of their target proteins [3]. Furthermore, since it is generally easier to determine the structures of small peptides with high accuracy than those of large proteins, conopeptides themselves can help determine the structures of the ion channels to which they bind with remarkable selectivity. For this reason, efforts have been devoted to characterizing the structures of a number of different conopeptides.

At the time of this article, there are 161 defined three-dimensional conopeptide structures in the PDB database [56], a number of which were refined with the help of docking and molecular dynamics. For example, MD simulations were used to refine the three-dimensional structures predicted from NMR data of conotoxin BtIIIA [105], α conotoxin MI [106], α conotoxin EIVA [107] and conantokin G in complex with calcium atoms [15]. Furthermore, by also using homology modeling based on a conotoxin with a known three-dimensional structure, ι-RXIA, Aguilar et al. [108] refined the structure of a new conotoxin, sr11a, a pore blocker in certain mammalian potassium channels. From a more ab initio perspective, Li et al. [109] and Platt et al. [110] used structure prediction algorithms followed by MD simulations to determine and assess three-dimensional structures for two different conopeptide systems, namely Vt3.1 and a subset of the conantokins, which can inhibit neuronal N-methyl-D-aspartate (NMDA) receptors in the brain.

As previously indicated, a number of computational studies have been performed demonstrating the use of conopeptides to aid in the structural determination of ion channels. For example, the comparison of a combined docking and MD simulation study with prior experimental data demonstrated the validity of an in silico approach for making microscopic predictions about complexes and the relevant microscopic forces controlling their interactions [111]. Molecular simulation refinement of a homology model of the complex of α conotoxin LvIA and the $\alpha_{3}\beta_{2}$ subtype of the nicotinic acetylcholine receptor (nAChR) based on the crystal structure of LvIA in complex with the acetylcholine-binding protein (AchBP) revealed important structural interactions between them [112]. Docking simulations of μ conotoxins demonstrated the validity of homology models for voltage-gated eukaryotic sodium channels, difficult to characterize experimentally, based on prokaryotic ones [113]. A more in-depth comparison of bacterial and mammalian sodium channel binding by μ-GIIIA, employing docking and biased and unbiased MD simulations, showed deeper insertion of the conotoxin into bacterial channels, which explained a corresponding loss of functionality of certain μ conotoxins when binding to the mammalian channels [114]. Finally, the α conotoxin family was used as a testing ground for a new docking algorithm called ToxDock, which employs ensemble docking to predict the structure of large receptor complexes bound to smaller peptide ligands [115].

#### 3.2.2. Molecular Mechanisms of Selectivity and Binding

Beyond shedding light on the detailed three-dimensional structures of conopeptides and receptor/toxin complexes, docking and MD have been employed to assess the relative importance of conopeptide structure, electrostatic distribution, and hydrophobicity to their interactions with ion channel receptors, as well as to identify the key residues within the structures that are responsible for the high pharmacological selectivity of conopeptides for specific receptor subtypes. For instance, MD and docking, both in conjunction with experimental data and from first principles, have shown how electrostatic effects contribute to pore-blocking the voltage-gated potassium and sodium channels and how they control selectivity for receptors. Specifically, docking studies were used to elucidate the importance of a positively-charged ring around the center of the conopeptides that bind to various subtypes of the voltage-gated potassium channels and similarly to elucidate the how certain basic residues of conopeptides interact with the negatively charged ring in the outer vestibule of different isoforms of the voltage-gated sodium channel [77,116,117]. In addition, to assess the contributions of various effects, Beissner et al. [118] used docking studies and MD simulations to calculate binding enthalpies to support their conclusion that charge is more important than steric contributions in controlling the selectivity of α conotoxins for the $\alpha_{3}\beta_{2}$ nAChR subtype over the $\alpha_{4}\beta_{2}$ subtype. Using a similar approach involving a computational scan and the calculation of binding energies for single point mutants of the α conotoxin ImI in complex with the $\alpha_{7}$ nAChR, Yu et al. [119] showed that dispersion and desolvation forces control its binding affinity to nAChRs, while electrostatic forces influence selectivity. Kwon et al. [120] employed MD simulations to probe the conformations and hydrogen-bonding networks of native and cyclic κ-PVIIA and explain the loss of interaction with the Shaker potassium channel occurring upon cyclicization, which they demonstrated to be primarily due to loss of electrostatic interactions with the N terminus, but also noted that certain hydrogen bonds formed by the native toxin might contribute to its stability. The microscopic forces that govern the changing electrostatics themselves have also been probed from an in silico perspective: Lúcio and Mazzoni [121] used a detailed quantum/classical approach to reveal structural and electronic changes occurring upon a single point mutation of ω-MVIIA and ω-MVIIC and to determine how changing electronic structure can predict changing hydrogen-bonding patterns; McDougal et al. [122] used constant pH MD simulations to probe the effects of pH upon the protonation state of key residues of α-MII.

Several studies have also identified overall hydrophobic and structural components to receptor/ligand interactions. Docking studies performed by Hopping et al. [123] rationalized, at a molecular level, the strong effect of hydrophobicity on selectivity of α- [A10L]PnIA for the $\alpha_{7}$ over the $\alpha_{3}\beta_{2}$ nAChR subtype; MD simulations performed by Cuny et al. [124] revealed that receptor side chain length is responsible for the experimentally determined affinities of α-RegIIA for human $\alpha_{3}\beta_{2}$ and $\alpha_{3}\beta_{4}$ nAChRs; and MD simulations performed by Chhabra et al. [125] demonstrated that a loss of key contacts of a *cis*-[2,8]-dicarba mutant of α-RgIA compared to the native form in complex with the nAChR $\alpha_{9}\alpha_{10}$ subtype was responsible for its lower binding affinities to the receptor. Pucci et al. [126] used a combination of MD and docking to elucidate the key interactions of the α-PIA N-terminal tripeptide tail with the $\alpha_{6}\beta_{2}$ nAChR subtype, while Lee et al. [127] employed a similar approach to demonstrate that the differences in the binding of α-GIC to the $\alpha_{3}\beta_{2}$ and $\alpha_{3}\beta_{4}$ nAChR subtypes are due to differing receptor side chain orientations (*cf.*
Figure 9 for an image of α-GIC in complex with a receptor).

Finally, a number of studies have probed the underlying molecular mechanisms of pharmacological selectivity via identification and analysis of the key residues involved in receptor/toxin binding, which often differ from peptide to peptide. Lin et al. [128] used homology modeling and docking studies to supplement analysis of the X-ray crystal structure of α-GIC with AchBP to show that His-5 in the conotoxin primarily contributes to binding to the $\alpha_{3}\beta_{2}$ and $\alpha_{3}\beta_{4}$ nAChR subtypes, whereas Gln-13 in the conotoxin primarily controls its selectivity for the $\alpha_{3}\beta_{2}$ subtype (see Figure 9). Kim and McIntosh [129] employed MD simulations as part of a demonstration that the mechanism of selectivity of α-BuIA for the $\alpha_{6}\beta_{2}$ nAChR subtype compared to the $\alpha_{4}\beta_{2}$ subtype is due to nonlocal interactions of the conotoxin with three key residues of the $\alpha_{6}\beta_{2}$ receptor: Lys-185, Thr-187, and Ile-188. Similarly, Kompella et al. [130] pinpointed a single residue difference between the rat and human nAChR $\alpha_{3}\beta_{2}$ subtypes–Glu-198 in the rat–that leads to lower binding affinity for α-RegIIA in the human receptor due to steric hindrance. Dutertre et al. [131] employed docking studies and binding energy calculations to verify their experimental hypothesis that the most important interaction controlling α conotoxin affinity for the $\alpha_{3}\beta_{2}$ nAChR subtype in general is a conserved proline interacting with residue Leu-119 in the receptor. Pérez et al. [132] used combined docking and MD to identify that Arg-7 and Arg-9 in α-RgIA control affinity and selectivity for the $\alpha_{9}\alpha_{10}$ nAChR subtype via interactions with Glu-195 and Asp-114 on the receptor. Grishin et al. [133] used MD and homology modeling to demonstrate that Phe-9 of α-AuIB determines key binding interactions of that toxin with the $\alpha_{3}\beta_{4}$ nAChR. Finally, Yu et al. [134] and Wu et al. [135] used positional scanning in α-TxID to demonstrate that Ser-9 is responsible for the selectivity of that conotoxin for the $\alpha_{3}\beta_{4}$ nAChR subtype; they then used MD simulations to identify the cause of the selectivity as attributable to minor steric changes in the binding pocket between different nAChR subtypes and to supplement design of a mutated analogue with greater affinity for $\alpha_{3}\beta_{4}$ due to putative disruption of a single hydrogen bond. In addition, a simulation and docking study identified a key methionine residue responsible for the toxicity of ω-MVIIA, information that may be particularly impactful as it has the potential to aid in studies designed to reduce the severe side effects in the analgesic ziconotide that limit its usefulness [136].

#### 3.2.3. Identification of Binding Sites, Complexes, and Pathways

In addition to probing the effects underlying conopeptide selectivity for receptor type and subtype, docking and molecular dynamics have been employed to characterize the actual binding modes: to identify binding sites and expected binding orientations and to qualitatively and quantitatively study binding and unbinding pathways.

As it is not always obvious what the actual binding site or orientation of binding of a ligand in a receptor is, and only a few receptor/toxin complexes have been crystallized, computational approaches have been invaluable in identifying and characterizing specific binding sites, as well as raising intriguing questions about the apparent multitargeting ability of certain conopeptides. For example, Ellison et al. [137] used docking studies of α-ImI to explain the respective affinities of it and α-ImII for different nAChR binding sites, and McArthur et al. [138] verified and explained the proposed binding orientation of μ-PIIIA in the voltage-gated sodium channel via experiment combined with MD simulation and Poisson-Boltzmann calculations. In addition, Cortez et al. [139] used docking studies to identify two different modes of binding of α-MI for binding sites on the α/δ interface of the *Torpedo marmorata* nAChR, and Grishin et al. [71] showed that two different structural isomers of α-AuIB bind to completely different sites on the $\alpha_{3}\beta_{4}$ nAChR subtype, despite both demonstrating affinity for the receptor and having identical sequences. Finally, Yu et al. [140] employed a combination of MD simulations and binding free energy calculations in conjunction with experiment to identify the binding site of α-Vc1.1 to the $\alpha_{9}\alpha_{10}$ nAChR subtype.

Once the binding sites have been identified, the dynamics of interaction between receptors and toxins can be examined, as has been done in several studies. Lin et al. [141] calculated binding free energies from MD to determine different metal-binding models for conantokin-T and conantokin-G, while Armishaw et al. [142] used docking refined with MD to investigate the binding interactions of α-ImI analogs produced from a large synthetic combinatorial library. Docking approaches have also been profitably employed by themselves: Luo et al. [143] elucidated the microscopic underpinnings of rapid unbinding of α-LtIA with the $\alpha_{3}\beta_{2}$ nAChR as part of the characterization of the overall complex and the identification of its novel binding mode and Dutertre et al. [144] highlighted the differences between conotoxin binding and snake toxin binding to the nAChR. Meanwhile Tietze et al. [145] performed an in-depth docking and MD analysis, including a model of the full toxin binding site, to probe the molecular basis of binding between δ-EVIA and voltage-gated sodium channels. Finally, Mahdavi and Kuyucak [146] undertook a systematic analysis of binding modes of different μ conotoxins to the Na${}_{V}$1.4 sodium channel subtype through both docking and MD, and Chen et al. [147] used a comprehensive combination of docking, nonequilibrium MD, and theoretical methods to calculate binding energies and predict the specificity of μ-PIIIA for eight different Na${}_{V}$ subtypes.

In addition to the fluctuations and interactions of the binding modes themselves, it is of interest to characterize binding and unbinding pathways by which conopeptides dynamically associate with their target receptors. Although these pathways are of clear importance in analyzing receptor/toxin interaction, they are somewhat difficult to characterize experimentally. A number of studies, primarily relying on nonequilibrium molecular dynamics approaches, have been performed to probe the kinetics of binding and unbinding. Two popular methods are thermodynamic integration [43] and umbrella sampling [148]. Often such studies are used to characterize free energy changes and potentials of mean force along the pathways. For instance, Chen and Chung [149] used such an approach to characterize multiple binding modes of ω-GVIA in the calcium Ca${}_{V}$2.2 channel subtype and accurately predicted IC50 values for channel inhibition. Chen et al. [150] performed umbrella sampling of the unbinding of μ-PIIIA from the Na${}_{V}$1.4 sodium channel and computed the potential of mean force and dissociation constants for the complex. Yu et al. [151] qualitatively reproduced the experimental binding affinities of α-ImI and α-PnIA [A10L,D14K] to AchBP and demonstrated that a large structural rearrangement of the receptor C-loop is necessary for unbinding, while Suresh and Hung [152] used umbrella sampling to compare the binding of α- [Y4E]GID with the $\alpha_{4}\beta_{2}$ nAChR subtype to that with the $\alpha_{7}$ subtype. In an example of the use of other nonequilibrium techniques, Yu et al. [153] employed random accelerated molecular dynamics and steered molecular dynamics to characterize multiple unbinding pathways of α-ImI from the $\alpha_{7}$ nAChR subtype and computed the potentials of mean force for unbinding using Jarzynski’s equation [154]. Meanwhile, Huang et al. [155] took a multiscaled approach by using a combination of atomistic molecular dynamics and coarse-grained Brownian dynamics simulations to model the binding process of κ-PVIIA: they were able to show that approach to the Shaker potassium channel is mediated by long-range electrostatics, while the final deep insertion is the result of a combination of electrostatic interactions from the Lys-7 side chain and hydrogen bonding and hydrophobicity primarily mediated by Phe-9 and Phe-23.

#### 3.2.4. Folding Kinetics and Isolated Conformations of Conopeptides in Solution

A slightly different but equally important question in the study of conopeptides is the study of their folding and dynamical conformations under different environmental conditions, an understanding of which provides a route forward for design of kinetically or thermodynamically controllable structures. Although fewer studies have been performed in this area, the number of such studies has increased in recent years. For example, Jiang and Ma [156] performed folding studies on α-GI with simplified quantum chemical computations and proved their usefulness in rapid simulations of folding/unfolding, while Karayiannis et al. [157] harvested trajectories from long parallel MD simulations of α-AuIB to quantitatively characterize its structural and dynamic properties: the fluctuations of its size and shape and its translational and rotational diffusivities in water. An interesting recent study done by Jain and Pirogova [158] probed the effects of electric field strength on the conformations taken on by MrIIIe in solution. Finally, Sajeevan and Roy performed MD simulations of α-AuIB and α-GI with disconnected disulfide bonds in water and water-ionic-liquid. They showed that the different solvents controlled the conformational landscape of the studied toxins, thus demonstrating that different ensembles of different isomers are thermodynamically favorable under different solvent conditions, potentially providing a route for thermodynamic control of in vitro folding [159,160].

As mentioned in Section 2.2, one of the strongest dictators of final conopeptide structure is disulfide connectivity, and several studies have been performed to probe aspects thereof, with somewhat peptide-dependent results. Paul George et al. [161] explored how the conformations of five different μ-conotoxins changed with successive removal of the native disulfide bonds. By doing so, they demonstrated that the set of conotoxins studied fall in a continuum from hirudin-like folding, in which folding proceeds by creation of intermediates with non-native disulfide connectivities, to BPTI-like folding, in which folding proceeds by progressive connection of native disulfide connectivities, and that the native structures of the studied toxins are retained upon removal of a single disulfide bond, but lost upon removal of two disulfide bonds. Several other simulations demonstrated that the removal of disulfide bonds is non-perturbative to the structure of cyclic conopeptides [162], but that non-cyclic conopeptides suffer greater structural perturbation [163]. Recently, Xu et al. [164] used MD simulations to show that all three disulfide bonds contribute significantly to the binding energies of μ-PIIIA to the Na${}_{V}$1.4 VGSC.

#### 3.2.5. Summary

We have compiled extensive docking and MD-based computational studies categorized under specific topics in a compact manner and summarized in Table A1. These studies have been invaluable in assessing the structures of conopeptides and predicting the structures of the ion channels to which they bind. They have further been of great use in characterizing the precise natures of the binding interactions between conopeptides and their target receptors. However, it is worth noting a few limitations associated with these computational studies. Only a narrow subset of the conopeptides have been explored with docking and MD studies, and the choice of which peptides to explore has been dictated more by interest in specific receptors or similarity to previously studied peptides than by any rational attempt to characterize the conopeptides as a class. Indeed, the majority of the studies have focused on a specific target, the nicotinic acetylcholine receptor. Furthermore, MD-based studies are time-consuming and computationally intensive and lack the high-throughput required for studying a large number of conopeptides. As we discussed in Section 2, out of the over 6000 identified sequences, less than a twentieth have associated characterized structures or targets, which may limit the usefulness of high-throughput analyses such as homology modeling and introduce difficulties in assessing overall trends. We recommend a few fruitful avenues of future inquiries. There is a need to have more iterative studies between experiments and theory that can help to improve the accuracy of computational methods and rationally inform experimental focus. Importantly, we envision the need to design further methodologies to integrate time-consuming, high-accuracy techniques such as ab initio MD with rapid, low-accuracy techniques such as homology modeling in a rational manner, which has the potential to lead to efficient, high-throughput studies that will result in greater understanding of the conopeptides as a whole. As we discuss in the next section, we are beginning to see a focus on novel, high-throughput computational techniques for design of conopeptides for specific applications, but further efforts are certainly called for.

### 3.3. Computational Design of Conopeptides for Specific Applications

In recent years, as the availability of computational resources has grown and confidence in computational algorithms has increased, there has been a concomitant increase in the number of design studies employing computational approaches in many areas, and the conopeptides are no different. For example, as part of a rational approach to design neurotensin analogues for pharmaceutical applications, Lee et al. [165] employed MD simulations and binding free energy calculations to demonstrate that glycosylation of contulakin-G lowers its affinity for the neurotensin 1 receptor, and as part of designing a methionine-lacking mutant, Ren et al. [166] employed MD simulations to demonstrate the effects of the methionine residue on α-TxID and its interactions with the $\alpha_{3}\beta_{4}$ nAChR subtype.

Perhaps more intriguingly, a number of more innovative, computationally-driven approaches have been introduced over the past decade. Through a combination of docking and MD simulations, Younis and Rashid [104] characterized the binding affinities of all available three-dimensional structures of conopeptides in the PDB database with a new target: the lysophosphatidic acid receptor 6 (LPAR6), which is implicated in several aggressive cancers. They identified α-BuIA as strongly binding to LPAR6, making it a good candidate for investigation and refinement as a possible anti-cancer drug. Gao et al. [167] performed a homology search of *Conus betulinus* venoms to identify six sequences similar to α-ImI, of which two were demonstrated to have desirable insecticidal properties, while Barba et al. [168] performed a sequence scan followed by MD simulations as the starting point for the design of an ω-GVIA mutant that strongly binds copper atoms to be used for environmental applications. Using conantokin-G as a starting sequence, Reyes-Guzman et al. [169] employed docking studies to evaluate mutants and guide a search that resulted in two peptides (EAR16 and EAR18) that are capable of reversibly blocking the GluN2B NMDA receptor, which is implicated in neuronal function. Of particular note have been several studies that have developed novel methodologies while also utilizing them for design purposes. In this vein, King et al. [170] used a genetic algorithm approach to perform a search through millions of sequences and designed a mutant of α-MII with more than double its binding affinity for the $\alpha_{3}\beta_{2}$ nAChR subtype. Two years later, the same authors used a more mature form of that approach in which they employed α-MII as a starting point for “drug repurposing”: searching a set of FDA-approved drugs for ones predicted to have high binding affinity to a new target, in this case once again the $\alpha_{3}\beta_{2}$ nAChR subtype [171]. Finally, Kasheverov et al. [172] employed a method they term Protein Surface Topography, which essentially creates a two-dimensional “topographical” map of the electrostatic potential of a ligand, to design an α conotoxin mutant with nanomolar affinity for the $\alpha_{7}$ nAChR subtype.

## 4. Future Outlook

As the study of proteins and peptides has matured, so too has the study of the conopeptides. A great deal of progress has been made on understanding their structures and functions, which in turn has shed valuable light on the structures and functions of ion channels and has improved methods for targeting those ion channels therapeutically. Computational approaches have helped to probe many different aspects of these inquiries: (i) machine learning predictors classify conopeptide targets with high accuracy on the basis of sequence; (ii) docking studies and molecular dynamics simulations reveal microscopic aspects of structure, binding, and dynamical conformations; and (iii) integrated computational approaches demonstrate their value for the rational design of conopeptides as therapeutic agents.

Many questions remain unanswered in the field of conopeptides. Identification of the three-dimensional structure of a new conopeptide is still a time-consuming process that often requires a combination of experimental techniques, such as X-ray crystallography and NMR, with docking and molecular dynamics, or computationally expensive ab initio folding. There are currently relatively few conopeptides for which pharmacological families have been determined, and, although classifiers with high accuracy exist, their generalizability to the full highly-diverse family of conopeptide sequences remains in question. Even fewer conopeptides are associated with verified three-dimensional structures, which makes knowledge-based prediction of targets or modes of action a difficult problem. In the remainder of this section, we discuss some of the challenges with overcoming these limitations.

The accurate prediction of the 3D structure of a protein from its sequence remains one of the “holy grails” of computational biology [39]. Ab initio (also called de novo) modeling approaches for obtaining protein structure predictions are very challenging and expensive except for small proteins such as “Trp-cage” (20 residues) [173], villin headpiece (35 residues) [174], NTL9 (39 residues) [175], and gpW (62 residues) [176]. Currently, ROSETTA is an actively used tool for de novo structure prediction from sequence, which is able to predict structures of single domain proteins up to a few hundred residues in length [177]. Structure prediction for a query sequence becomes more tractable when related sequences have available experimentally-resolved structures; this is referred to as homology modeling [178]. For typical proteins (at least 100 amino acids long), a general rule for building a homology model of a protein with unknown structure using another protein as the structural template is that both proteins should share at least 25% sequence identity; below this is considered an uncertain “twilight zone” [179,180,181,182]. Homology modeling has already been applied in a few cases to predict conopeptide three-dimensional structures [108,183,184,185]; however, applying it to an arbitrary conopeptide still suffers from notable difficulties. The majority of conopeptides do not reach the “typical” protein length of 100 amino acids and would therefore require a higher sequence identity cutoff for identification of an appropriate template for homology modeling, which in turn implies that such a template may not exist due to the high sequence variability of the mature toxins and the small number of determined three-dimensional structures [185]. Although their short lengths makes them more tractable for ab initio approaches, the large number of conopeptide sequences renders this approach of limited use for the characterization of the family itself.

As discussed in Section 2.2, the three-dimensional structure of the cysteine-rich peptides that comprise the bulk of the known conopeptides is largely determined by their disulfide connectivity. Indeed, in the few cases where homology modeling of conopeptides has been successfully applied, the appropriate disulfide connectivity for cysteine-containing conopeptides was known or assumed a priori [108,183,184]. In theory, prediction of the connectivity should allow the construction of a reasonable three-dimensional structure for any cysteine-rich conopeptide by analogy with another three-dimensional structure harboring the same connectivity. In practice, however, this is not straightforward. The number of possible connectivities grows extremely rapidly: in a protein with *n* cysteines the total number of possible connectivities is [186],
(1)$$\frac{n!}{\left(\frac{n}{2}\right)!2^{n/2}},$$
which is a tractable three connectivities for a four-cysteine framework, but grows to fifteen for a six-cysteine framework, and over a hundred for an eight-cysteine framework. Indeed, the prediction of disulfide connectivity in general is still an active area of research, although state-of-the-art methods are beginning to reach over 80% accuracy in general [187,188,189,190].

Although most currently characterized conopeptides with a particular cysteine framework are claimed to have the same native disulfide connectivity, it is not obvious to what extent sequence similarity among the relatively small subset of conopeptides with known three-dimensional structures is responsible for this, which calls into question the generalizability of this trend. More intriguingly, there is evidence that different connectivities, observed experimentally, can bind to different receptor subtypes with reasonable affinity [71] and that different connectivities can represent metastable, kinetically-trapped states under certain conditions [160,191]. Some evidence even points to the existence of a thermodynamic ensemble of different structural isomers that can be affected by environmental factors [161]. Even if only one native isomer exists in vivo, kinetic rather than thermodynamic control in vitro could be a possible avenue for design of conopeptides with novel action. In vitro studies often have difficulty in correctly forming “native” disulfide connectivities [192], but from the perspective of isomeric design this might be turned into an advantage.

## 5. Conclusions

The series of venoms produced by the family of cone snails provides a diverse wealth of short disulfide-rich peptides with potential pharmaceutical value due to their high affinities for specific ion channel receptors. In this review, we have discussed the various efforts that have been made from a computational perspective to classify and characterize many of the different conopeptides. We have described the difficulties associated with drawing general conclusions about a set of such short, sequentially diverse peptides in the context of potential areas of future research. As the impact of computational approaches continues to grow, we hope to see an increase in the number of large-scale and high-throughput computational studies of the conopeptides, supported by a continuous increase in available structural data from experiment, with a concomitant increase in their value as novel ion-channel targeting drugs.

## Figures and Tables

**Figure 1 marinedrugs-17-00145-f001:**
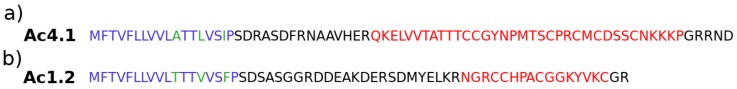
Comparison of two conotoxin sequences from Gene Superfamily A. Matching precursor signal sequence regions are in blue, places where the precursor regions do not match are in green, and the mature toxin regions are in red. The remainder of the sequences comprising the N-terminal and C-terminal pro-regions are in black. (**a**) sequence of conotoxin Ac4.1 from *Conus achatinus*; (**b**) sequence of conotoxin Ac1.2 from *Conus achatinus*.

**Figure 2 marinedrugs-17-00145-f002:**
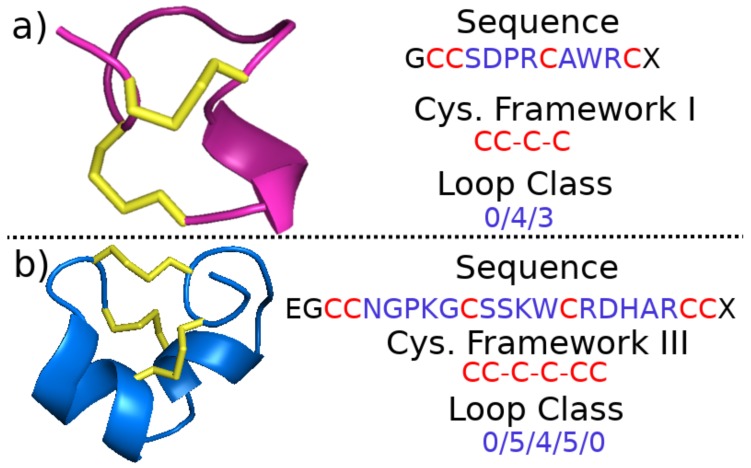
Illustration of sequence, cysteine framework and loop class for (**a**) α-conotoxin ImI (Protein Data Bank or PDB structure 1G2G [51]) and (**b**) μ-conotoxin CnIIIC (PDB structure 2YEN [52]). In the illustrative 3D structures on the left, disulfide bonds are represented as yellow sticks. Images of peptides were generated with Pymol [53].

**Figure 3 marinedrugs-17-00145-f003:**
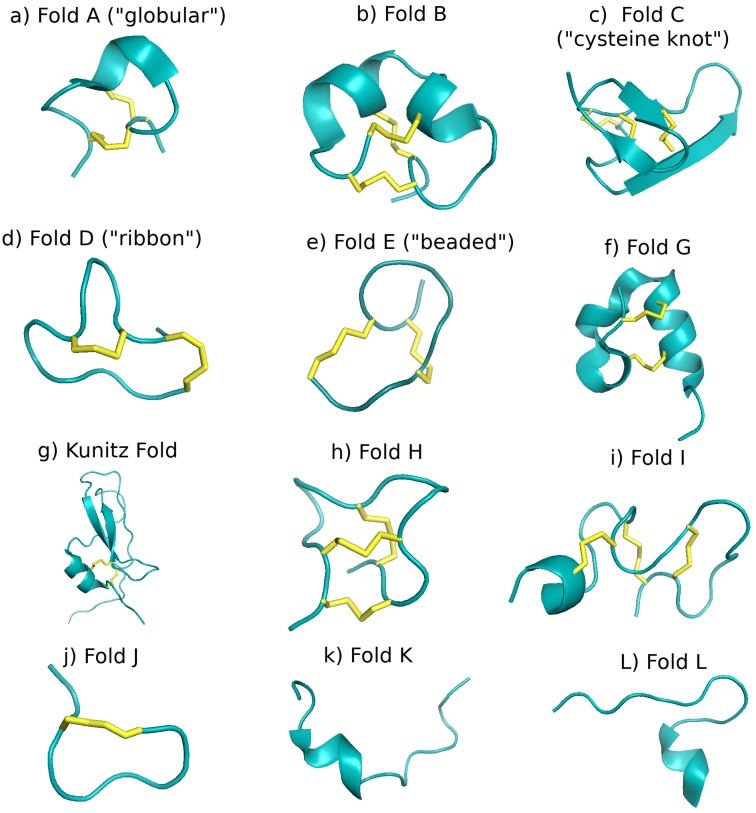
The thirteen major conopeptide folds described in Akondi et al. [3], shown by representative examples from the PDB [56]. Disulfide bonds are represented as yellow sticks. All images rendered in Pymol [53]. (**a**) Fold A, also referred to as the “globular” fold: conotoxin α-RgIA, PDB structure 2JUT [57]; (**b**) Fold B: conotoxin μ-CnIIIC, PDB structure 2YEN [58]; (**c**) Fold C, also referred to as “cysteine knot” fold: conotoxin δ-EVIA, PDB structure 1G1P [59]; (**d**) Fold D, also referred to as “ribbon” fold: ribbon isoform of conotoxin α-GI, PDB structure 1XGB [60]; (**e**) Fold E, also referred to as “beaded” or “beads-on-a-string” fold: conotoxin χ-CMrVIA, PDB structure 2B5Q [61]; (**f**) Fold G: conotoxin κ-PIXIVA, PDB structure 2FQC [62]; (**g**) Kunitz fold: conkunitzin-S2, PDB structure 2J6D [63]; (**h**) Fold H: conotoxin MrIIIe, PDB structure 2EFZ [64]; (**i**) Fold I: Conotoxin α-PIVA, PDB structure 1P1P [65]; (**j**) Fold J: contryphan-Vn, PDB structure 1NXN [66]; (**k**) Fold K: conantokin-G, PDB structure 1AD7 [15]; and (**l**) Fold L: conomarphin, PDB structure 2YYF [67]. There is no representative structure in the PDB for Fold F. The interested reader is referred to the original paper by Zhang et al. [68], which contains the characterization of the only determined structure of this fold.

**Figure 4 marinedrugs-17-00145-f004:**
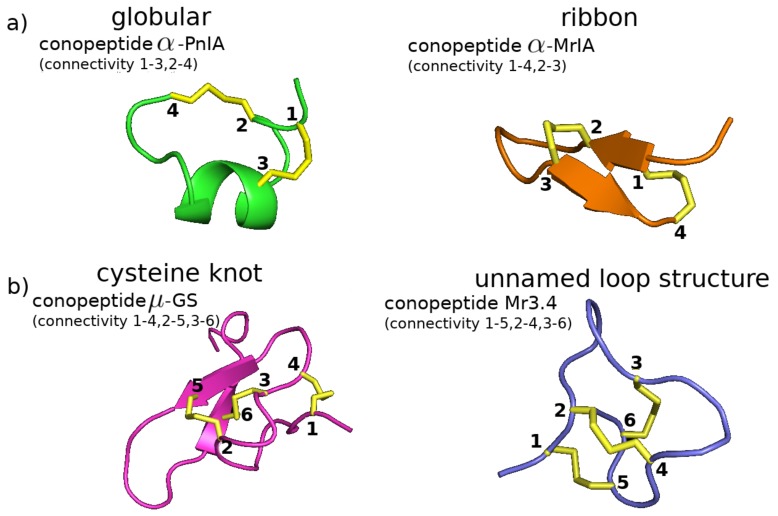
Different disulfide connectivities lead to different conotoxin structures. (**a**) With four cysteines, two different connectivities can lead to either a “globular” structure with α-helical content (left, PDB 1PEN for conotoxin α-PnIA [72]) or a flattened “ribbon” structure which often, but not always, displays β-sheet content (right, PDB 2EW4 for conotoxin α-MrIA [73]). (**b**) With six cysteine residues, two connectivities that differ only in the first two disulfide bonds can lead to either a “cysteine knot” structure with β-sheet content (left, PDB 1AG7 for conotoxin μ-GS [74]) or another structure with no discernable secondary structure content (right, PDB 2EFZ for conotoxin MrIIIe [64]).

**Figure 5 marinedrugs-17-00145-f005:**
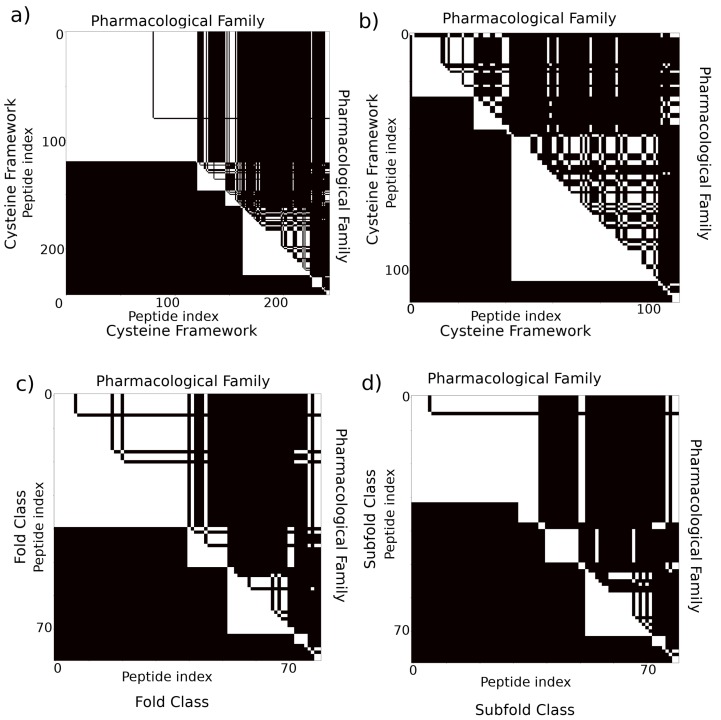
Comparison of different categories with pharmacological family. (**a**) Comparison between cysteine framework and pharmacological target for all pairs of 243 peptides with determined pharmacological targets downloaded from the Conoserver [17,54]. Black indicates that two peptides are assigned different categories, while white indicates the two peptides are assigned the same category. The lower triangular shows cysteine framework; the upper triangular pharmacological target. (**b**) Comparison between cysteine framework and pharmacological target for the subset of 106 sequences with more than four cysteines. In (**c**) and (**d**), we show a comparison between fold and subfold class and pharmacological family for all pairs of 80 peptides with a defined pharmacological target and fold/subfold classes as described in Akondi et al. [3].

**Figure 6 marinedrugs-17-00145-f006:**
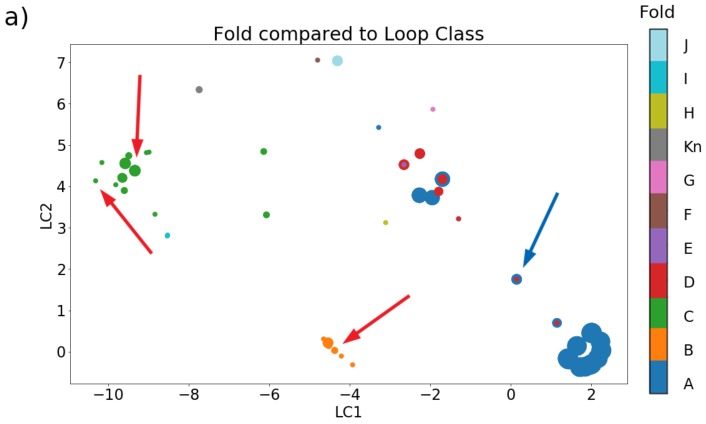
Two-dimensional embedding of loop class, demonstrating its relationship to (**a**) fold and (**b**) pharmacological family. Data compiled from Akondi et al. [3], which includes 103 peptides with measured structures, 80 of which had identified pharmacological targets. Loop class was represented as a seven-dimensional vector, with vectors representing classes containing fewer than eight cysteines padded with negative ones for direct comparison. In all images, size of a given marker indicates the number of conopeptides with identical loop class and category, while color indicates category. Red arrows draw the reader’s attention to differences in clustering between the two panels. Blue arrows indicate an example of a structural isomer. The embedding was done for visualization purposes employing the t-Distributed Stochastic Neighbor Embedding (t-SNE) algorithm as implemented in Scikit Learn [79,80].

**Figure 7 marinedrugs-17-00145-f007:**
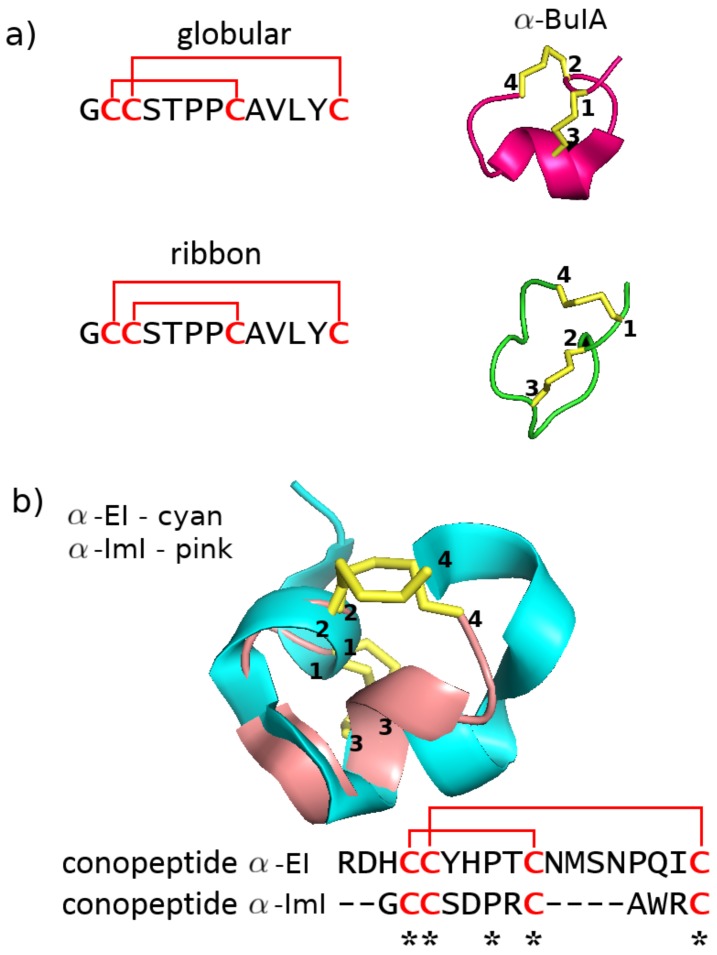
Impact of disulfide connectivity on conotoxin structure. (**a**) The same sequence can adopt different structural folds given different disulfide connectivities [69]. Conotoxin α-BuIA in globular form with connectivity 1-3, 2-4 (top, PDB structure 2I28 [82]) and in ribbon form with connectivity 1-4, 2-3 (bottom, PDB structure 2NS3 [83]). (**b**) Different sequences can adopt similar structural folds given the same disulfide connectivity. Conotoxin α-EI in cyan (PDB structure 1K64 [84]) and conotoxin α-IMI in pink (PDB structure 1G2G [51]), both in globular form with connectivity 1-3, 2-4. All images rendered in Pymol with disulfide bonds represented as yellow sticks [53].

**Figure 8 marinedrugs-17-00145-f008:**
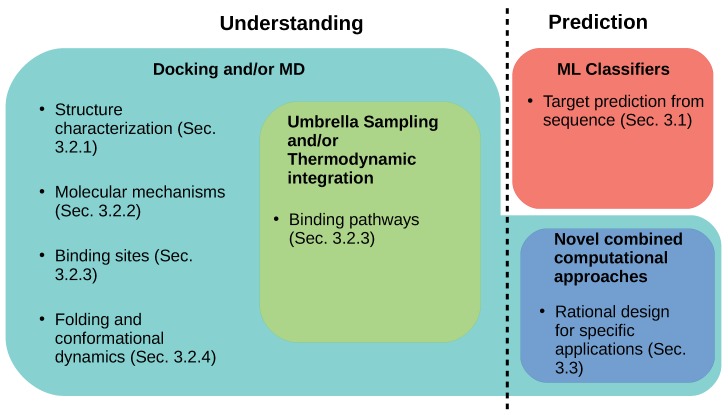
A visual overview of Section 3, with a focus on the different computational techniques employed.

**Figure 9 marinedrugs-17-00145-f009:**
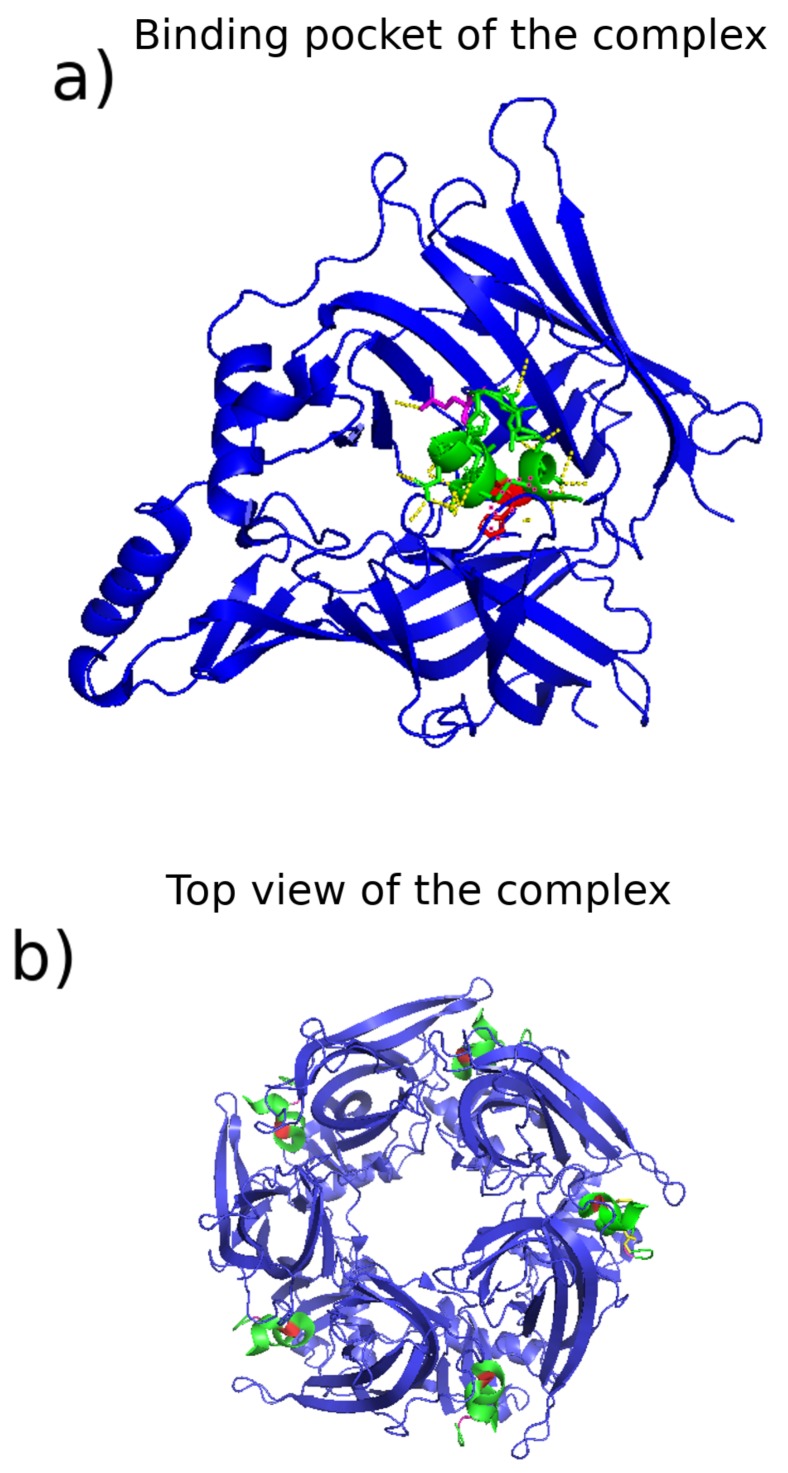
Illustration of binding modes and key residues. Figure shows conotoxin α-GIC in complex with the acetylcholine binding protein (AchBP) from *Aplysia californica*, often employed for homology modeling of nAChRs. Structure downloaded from the PDB server (code 5CO5). We highlight the key residues for selectivity identified by the study that also characterized the crystal structure through a combination of experimental and computational techniques [128]—in red, His-5, and in magenta, Gln-13, shown to control binding affinity and selectivity, respectively, for the $\alpha_{3}\beta_{2}$ nACHr subtype. (**a**) shows a zoom in of one of the binding pockets, with hydrogen bonds represented as yellow dotted lines, while (**b**) shows a top view. In both panels, the main body of the conotoxins are colored green and AchBP is colored blue. Images rendered with Pymol [53].

**Table 1 marinedrugs-17-00145-t001:** Different categories used to classify the conopeptides, along with the basic type of categorization and a brief description.

Category	Type	Description
Gene superfamily	sequence	Clustering of precursor region
Cysteine framework	sequence	Arrangement of cysteines
Loop class	sequence	Number of amino acids between cysteines
Disulfide connectivity	structure	Pattern of disulfide bond formation
Fold	structure	General three-dimensional structure
Subfold	structure	More specific three-dimensional structure
Pharmacological family	action	Target and mode of action (agonist, antagonist, etc.)

**Table 2 marinedrugs-17-00145-t002:** Summary of cysteine frameworks, with defining pattern and number of cysteines. Data compiled from the Conoserver, an automatically-updating online repository of conopeptide data [17,54]. In the entry for framework X, .[PO] represents an interceding hydroxyproline residue.

Framework Name	Pattern	No. Cysteines
I	CC-C-C	4
II	CCC-C-C-C	6
III	CC-C-C-CC	6
IV	CC-C-C-C-C	6
V	CC-CC	4
VI/VII	C-C-CC-C-C	6
VIII	C-C-C-C-C-C-C-C-C-C	10
IX	C-C-C-C-C-C	6
X	CC-C.[PO]C	4
XI	C-C-CC-CC-C-C	8
XII	C-C-C-C-CC-C-C	8
XIII	C-C-C-CC-C-C-C	8
XIV	C-C-C-C	4
XV	C-C-CC-C-C-C-C	8
XVI	C-C-CC	4
XVII	C-C-CC-C-CC-C	8
XVIII	C-C-CC-CC	6
XIX	C-C-C-CCC-C-C-C-C	10
XX	C-CC-C-CC-C-C-C-C	10
XXI	CC-C-C-C-CC-C-C-C	10
XXII	C-C-C-C-C-C-C-C	8
XXIII	C-C-C-CC-C	6
XXIV	C-CC-C	4
XXV	C-C-C-C-CC	6
XXVI	C-C-C-C-CC-CC	8
XXVII	C-CC-C-C-C	6

**Table 3 marinedrugs-17-00145-t003:** Summary of pharmacological families, their targets, and their modes of action. Data compiled from the Conoserver [17,54] and the Uniprot database [75].

Family	Target	Mode of Action
α (alpha)	Nicotinic acetylcholine receptors (nAChRs)	orthosteric, allosteric inhibition
γ (gamma)	Neuronal pacemaker cation currents	increase calcium current
δ (delta)	Voltage-gated sodium channels (VGSCs)	agonist, delayed inactivation
ϵ (epsilon)	Presynaptic calcium channels or G protein-coupled presynaptic receptors (GPCRs)	blocker
ι (iota)	VGSC	agonist, no delayed inactivation
κ (kappa)	Voltage-gated potassium channels (VGPCs)	blocker
μ (mu)	VGSC	antagonist, blocker
ρ (rho)	Alpha-1 adrenergic receptors	allosteric inhibitor
σ (sigma)	Serotonin-gated ion channels	antagonist
τ (tau)	Somatostatin receptor	antagonist
χ (chi)	Neuronal noradrenaline transporter	unknown
ω (omega)	Voltage-gated calcium channels (VGCCs)	blocker

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
