# Peer review of "Snails In Silico: A Review of Computational Studies on the Conopeptides"

_marinedrugs, 2019, doi:10.3390/md17030145_

Reviewer 1 Report

This review describes the computational studies of conotoxins. The review starts with a description of conotoxins classification schemes, the describe machine learning methods for predicting activity from sequence, then the docking/molecular dynamics studies are described. The review is mostly well written and interesting. I used the line numbering as provided in the downloaded manuscript.

GENERAL

Comment 1.  The description of the docking studies is a collection of conclusions from individual articles rather than a summary/comparison/critics of all these results.

Comment 2. The figures could be improved. For example, Figure 9 is not very useful and it would probably be better to represent binding modes extracted from the papers that were cited (asking permission from corresponding editors). As a side note, the secondary structure of RgIA (panel b) needs to be recomputed within Pymol to display the same representation as the other toxins. (the command is 'util.ss').

INCORRECT STATEMENTS

Comment 3: The authors are confusing "conopeptide" and "conotoxin". Conopeptides describe all peptides found in the venom of cone snails. Conotoxins are all the disulfide-rich conopeptides, and are therefore a subset of the conopeptides. There are therefore not interchangeable terms. In the literature, it is describe that there was ~1,000 conopeptides (and not conotoxins) in a cone snail venom (Line 23).

Comment 4. Line 1: "Marine cone snails are carnivorous gastropods that use paralytic peptide toxins called conotoxins both as a defense mechanism and as a means to immobilize and kill their prey.". Not all conotoxins are paralytic, some are excitatory, some have other activity.

Comment 5. Line 31: "other types of modifications have also been observed, including proline hydroxylation [11], tyrosine sulfation [12], C-terminal amidation [13], and O-glycosylation [14]." Please cite also the gamma-carboxyglutamic acid, as they are frequently found. Also write "such as" instead of "include" as many other modifications have been found and not described here.

Comment 6. Line 379, states that cyclic PVIIA had decrease in activity compare to the linear peptide because of a change of h-bond network. This is not correct, the article describes that PVIIA establishes a charge interaction with its N-terminus, and the loss of this interaction after backbone cyclization was suggested to be responsible for the loss of activity.

Comment 7. Line 457, "Often such studies are used in conjunction with thermodynamic integration to characterize free energy changes and potentials of mean force along the pathways." Umbrella sampling is a different technique from thermodynamic integration. Umbrella sampling uses the WHAM algorithm to build the potential of mean force.

Comment 8. Line 465, "In addition to umbrella sampling, Yu et al. [147] employed random accelerated molecular dynamics and steered molecular dynamics to characterize multiple unbinding pathways of α-ImI from the α 7 nAChR subtype and once again computed the potentials of mean force for unbinding." This is not correct. The cited reference used RAMD to study the unbinding pathway and then steered MD long these pathways to compute the potential of mean force using the Jarzynski equation. There was no umbrella sampling involved in that work.

CONFUSING STATEMENTS

Comment 9. Line 15: "We close with a discussion of open questions in the field." This sentence is confusing.

Comment 10. Line 413. "a particularly intriguing simulation", why is this simulation "intriguing"?

OTHER

Comment 11. A number of references have issues with duplication of names (for example the first one, but there are others).

Author Response

We are delighted that the reviewer finds our work to be well-written, interesting, and a significant contribution to the field. We thank the reviewer for their time and effort in assembling such a detailed and helpful critique. In the attached document, we address the specific issues raised in the review, and detail the changes we have made within our revised submission. We believe that the modifications we have made have strengthened the clarity and precision of our work.  Where possible, we have highlighted changes in red in the revised manuscript, as well as noting them by line number and highlighting specific wording changes in blue in this response.

Reviewer 2 Report

The review of Mansbach et al. focuses on computational studies where conopeptides have been involved. Today it is one of the few reviews on this topic and the presence of more than 180 references makes it very useful for the interested reader. Other advantages of the review include its good and reasonable structure and, in particular, the availability of a summary Table A1 with all listed conopeptides.

After reviewing, I have had just a few questions and suggestions that may be useful to improve this manuscript.

1. The review does not contain a figure or at least description of 12 conopeptide folds (section 2.1.3.), and is only a reference to the work of Akondi et al. [3]. And this is despite the fact that some of these folds are actively used in Figures 4c, d and 5a. It would be important to realize what fold is in consideration without finding an additional citation. My recommendation is an extra figure with conopeptide folds (all or the most common).

2. In addition, the reviewing caused an unfortunate feeling that in some cases the authors interpret the results of the cited sources rather freely. Here are some examples:

- (Section 3.2.1., last paragraph, lines 341-342). In work [101] "a crystal structure of the complex of a conotoxin LvIA and the a3b2 subtype of the nicotinic acetylcholine receptor (nAChR)" is absent, as the reader might think. Contrary to the text, the caption to Figure 8 correctly states that the crystal structure of this peptide was obtained with acetylcholine-binding protein, not a receptor.

- A similar inaccuracy occurs in the next section 3.2.2. (line 394):"to supplement analysis of the X-ray crystal structure of a-GIC with the a3b2 and a3b4 nAChR subtypes". We are talking here again about the crystal structure of conotoxin with acetylcholine-binding protein, not receptors.

- One more example of careless attitude to the source is in the last paragraph of section 3.3 (lines 535-536): "which essentially creates a two-dimensional "topographic" map of the electrostatic potential of a target and its proposed agonist". The method proposed by the authors in this paper did not involve obtaining a topographic map of the target, but only a ligand (conotoxin), which, by the way, is not an agonist, but a competitive antagonist on the a7 nAChR subtype. Such an arbitrary interpretation of the mode of action of the conopeptides on their targets can cause confusion among readers. Why, for instance, in Table 3 for the alpha-family (which includes alpha-conotoxins) the mode of action is specified as "allosteric inhibitor"? For a large number of different alpha-conotoxins was shown their inhibitory function through the nAChR orthosteric binding sites.

Author Response

We are greatly pleased that the reviewer finds our work to be useful and well-structured. We thank the reviewer for the time and effort they have spent in assembling a thorough and helpful critique. In the attached document, we address the specific issues raised in the review, and detail the changes we have made within our revised submission. We believe that the modifications we have made have strengthened the clarity and precision of our work

Reviewer 3 Report

Mansbach and colleagues have provided an extensive and well-written review on the state of the art on computational studies of conotoxins. This work is valuable contribution to the field. I only have a minor comment:

A recent work (Discovery of Novel Conotoxin Candidates Using Machine Learning. Li Q, Watkins M, Robinson SD, Safavi-Hemami H, Yandell M. Toxins (Basel). 2018 Dec 1;10(12). pii: E503. doi: 10.3390/toxins10120503) has introduced ConusPipe, a machine learning tool that can be used to predict whether a certain transcript in a Conus transcriptome is a putative conotoxin. I believe this tool should be discussed in this review.

Author Response

Mansbach and colleagues have provided an extensive and well-written review on the state of the art on computational studies of conotoxins. This work is valuable contribution to the field. I only have a minor comment:
A recent work (Discovery of Novel Conotoxin Candidates Using Machine Learning. Li Q, Watkins M, Robinson SD, Safavi-Hemami H, Yandell M. Toxins (Basel). 2018 Dec 1;10(12). pii: E503. doi: 10.3390/toxins10120503) has introduced ConusPipe, a machine learning tool that can be used to predict whether a certain transcript in a Conus transcriptome is a putative conotoxin. I believe this tool should be discussed in this review.

Author response: We thank the reviewer for their kind words: we are grateful that they find our work well-written and valuable. We further thank them for bringing this recent work to our attention, as we agree that it warrants inclusion in our review. As such, we have added the following to our review:

Page 14, Lines 268-271: “Additionally, a very recent ML tool, ConusPipe, has attempted to classification of RNA sequences through three different models, in order to identify potential novel conotoxin sequences within Conus transcriptomes without resorting to sequence homology type searches [96].”

Page 24, Table A1, Entry 5:

Toxin(s): RNA sequences from ten species

Methods: Logit, Label spreading, Perceptron

Results/Citations: ConusPipe identifies potential conotoxins from sequence [96]